# Identifying Loss of Control Eating within Childhood Obesity: The Importance of Family Environment and Child Psychological Distress

**DOI:** 10.3390/children7110225

**Published:** 2020-11-11

**Authors:** Ana Rosa Sepúlveda, Tatiana Lacruz, Santos Solano, Miriam Blanco, Alba Moreno, Marta Rojo, Lucía Beltrán, Montserrat Graell

**Affiliations:** 1Department of Biological and Health Psychology, School of Psychology, Autonomous University of Madrid, 28049 Madrid, Spain; Tatiana.lacruz@gmail.com (T.L.); santos.solano@uam.es (S.S.); miriam.blancoh@uam.es (M.B.); alba.moreno@uam.es (A.M.); marta.rojo@uam.es (M.R.); lucia.beltran@uam.es (L.B.); 2Department of Child and Adolescent Psychiatry and Psychology, University Hospital Niño Jesús, 28009 Madrid, Spain; montserrat.graell@salud.madrid.org

**Keywords:** childhood obesity, eating disorders, loss of control eating, expressed emotion, family

## Abstract

This study aims to examine the differences in family environment, psychological distress, and disordered eating symptomatology between children classified by weight status with or without loss of control (LOC) eating and to test a model of the role of emotional regulation of LOC eating based on a dysfunctional family environment. A cross-sectional study was conducted among 239 families. The assessment measured family expressed emotion, family adaptability and cohesion, child levels of depression and anxiety, body esteem, and disordered eating attitudes. The assessment was carried out in primary care centers and primary schools. Child body mass index (BMI) was associated with higher expressed emotion, psychological distress, and disordered eating symptomatology. Children with obesity and LOC presented higher BMI, poorer body esteem, and more disordered eating attitudes than children without LOC. Children with overweight/obesity, both with or without LOC, exhibited higher psychological distress and emotional overinvolvement than normal-weight children. A partial mediation of depression or anxiety and disordered eating attitudes between expressed emotion and LOC was found. Findings support that children with overweight/obesity show more family and psychological distress. Body esteem issues and disordered eating attitudes could alert the presence of LOC in children with obesity. The function of LOC might be to cope with psychological distress that may appear in a dysfunctional family environment.

## 1. Introduction

Binge eating disorder (BED) is one of the eating disorders that is more frequently associated with adult obesity [1]. Binge eating is defined as eating an objectively large amount of food accompanied by the experience of loss of control (LOC) overeating [2]. However, determining what constitutes an “objectively large amount of food” in growing children with varying nutritional needs can often be tricky and is challenging for behavior screening [3]. In this manner, researchers studying disordered eating in children have begun to rely more on the behavior of LOC eating.

LOC eating has been defined as “eating with the associated experience of being unable to control how much one is eating independent of the amount of food consumed” [4]. This definition naturally allows for the inclusion of both objective and subjective binge eating episodes. In a recent meta-analysis [5], the authors concluded that more than one-third of children with overweight and obesity presented LOC eating. The affect regulation model suggests that this behavior is as a result of maladaptive coping with negative emotions [6], which in some cases seems to be related to difficulties in solving interpersonal problems [7].

Furthermore, in a recent critical review of the literature, Goldschmidt (2017) also concluded that LOC eating was a psychopathology construct that appears to be uniquely associated with psychological distress and impairment, disturbed eating behaviors, and weight-related factors in both cross-sectional [8] and prospective studies [9,10]. A series of published meta-analyses proposed that the experience of LOC during eating was a salient marker of disordered eating [3,5,11].

### 1.1. LOC Eating Onset and Associated Family Factors

A dysfunctional family environment is considered a risk factor of child psychological distress [12], eating disorders [13], and childhood obesity [14]. Specifically, a lack of emotional support and critical family comments about weight, shape, and diet are a shared risk factor in eating disorders [13], obesity [15], and LOC eating [16].

In this way, another relevant aspect of family environment is expressed emotion (EE), which is frequently measured in eating disorders [13]. EE is a construct that can be described as negative attitudes, in the form of criticism, hostility, and emotional overinvolvement, shown by family members toward a child with a psychological problem. It may reflect the quality of family relationships, which can affect the physical and emotional wellbeing of a child. A high EE family environment can be considered a maintenance stressor that may interact with individual biological vulnerabilities to promote illness onset and recurrence [12]. This construct, however, has been scarcely used in obesity research [17]. However, there is an ample body of research that supports EE as a predictor for the outcome of several psychiatric disorders [12], including binge eating [18]. Since other factors, such as family conflict [19], have been linked to the development of LOC episodes, LOC eating and binge eating may arguably share the predictive value of this construct.

### 1.2. Identifying LOC Eating in Childhood Obesity

The relationship between obesity and eating disorders is considered bidirectional, so the presence of one of them is a risk factor for developing the other [20]. In the treatment of obesity, it is recommended to initially detect the presence of psychological comorbidity that may interfere with the adaptation and evolution of treatment, especially in the case of eating disorders [1].

However, the role of LOC in childhood obesity is yet to be fully understood. Some studies refer to childhood obesity and LOC eating as different types of disordered eating behavior [21], while others assess LOC eating within the population with childhood obesity [22] and others compare it with the general population [8,23,24].

Childhood obesity, regardless of the possible presence of LOC eating, has also been positively associated with the presence of psychological problems [25] and a dysfunctional family environment [14,26].

Ideally, professionals dedicated to the treatment of childhood obesity should be able to discriminate the presence of LOC eating in their patients to carry out effective treatment. A recent review [3] highlights that thoroughly understanding the risk factors for the development of LOC eating in childhood through adulthood is crucial to the development of effective prevention and treatment efforts.

With this in mind, the underlying objectives of the current study were: (a) to describe the family environment (EE and family adaptability and cohesion); psychological distress (depression and anxiety); and disordered eating symptomatology (body esteem and disordered eating attitudes) in three child samples by weight status (normal-weight [NG], overweight [OWG], and obese [OG]) and the relationship between child body mass index (BMI) and these variables; (b) to examine the differences in family environment, psychological distress, and disordered eating symptomatology between combined weight status groups with LOC eating and without, comparing with an NG without LOC eating; and (c) to test a model of emotional regulation of LOC eating based on a dysfunctional family environment. The hypotheses are the following: (a) BMI will be associated with higher levels of EE, lower levels of family functioning, higher levels of psychological distress, and higher levels of disordered eating symptomatology; (b) LOC eating episodes will be more frequent in children with OWG/OG than in NG and children with LOC eating will be associated with a more highly dysfunctional environment, higher psychological distress, and higher disordered eating symptomatology than in children without LOC eating; and (c) anxiety and depression, evaluated separately in two models, will mediate the relationship between dysfunctional family environments and LOC eating episodes. Furthermore, disordered eating symptomatology was used as a secondary mediator.

## 2. Materials & Methods

### 2.1. Participants and Procedure

The sample was recruited from the ANOBAS study (PSI2011-23127), a cross-sectional case-control study designed to assess early risk factors for childhood obesity. The sample consisted of 239 Spanish families of children aged 8 to 12 (Mean [M] = 10.35; standard deviation [SD] = 1.45). The sample consisted of 31.7% normal-weight (n = 76), 33.8% overweight (n = 81), and 34.3% obese (n = 82) children, according to age- and sex-specific cut-off points [27], in which 57.7% were females and 42.3% were males. The family’s socioeconomic status (SES) was calculated according to Hollingshead’s index, resulting in 9.1% as low, 16.9% as medium-low, 35.5% as medium, 18.2% as medium-high, and 20.3% as high.

The OWG and OG sample were recruited from different Primary Care Centers in Madrid through response flyers provided by their child’s pediatrician at a routine checkup. The normal-weight sample (15th–85th percentile) was recruited from several primary schools (all in geographical proximity). Details of the recruitment and sample selection procedure have been published elsewhere [28,29]. Those participants who presented secondary obesity (e.g., due to Cushing Syndrome) and who did not have an adequate oral or written command of Spanish, who suffered from a developmental disorder (e.g., autism spectrum disorders or intellectual disability) were excluded from the study.

All of the participants were recruited over a period of four years (between 2012 and 2016). The ANOBAS study received ethical approval by the Niño Jesus Children’s Hospital (No. Ref. 0009/10), Primary Care Commission (Ref. 11/12), and the corresponding university committee (UAM, Research ethics committee 27-673/76-1394). Participation was voluntary and participants did not receive remuneration. Participating families signed an informed assent and consent form.

Participants were summoned to their Primary Care Center or their primary school, where a trained interviewer completed a semi-structured interview, with the primary caregiver and the child separately, and provided the questionnaires. Family environment questionnaires were answered by the primary caregiver, while the remaining questionnaires were answered by the child. Both the primary caregiver and the child answered the questionnaires alone and handed them over to the interviewer at the time of the assessment.

### 2.2. Instruments of Assessment

Children’s anthropometric data. Height and weight were measured using a digitally calibrated scale (Type SECA 799 and 769) and a tallimeter.

Family environment. Family environment was assessed using two measures (EE and patterns of family functioning). EE was assessed using the Family Questionnaire (FQ) [30]. It consists of 20 items: 10 for the criticism scale (CC) and 10 for the emotional over-involvement scale (EOI). High expressed emotion (High-EE) was categorized by a score above any of these cut-off points (CC’s scale ≥ 23 score or EOI’s scale ≥ 27 score). The Cronbach’s alpha for the CC subscale is 0.83 and 0.72 for the EOI subscale for the Spanish version [30]. For the present study, coefficients were slightly higher (CC α = 0.85 and EOI, α = 0.76).

Patterns of family functioning were assessed using the Family Adaptation and Cohesion Scales (FACES-III) [31]. This instrument consists of 20 items distributed in two scales: Adaptability and Cohesion. The Spanish version of the scale presents adequate psychometric properties (α = 0.72 and α = 0.68 for the Cohesion and Adaptability scales, respectively). For the present study, Cronbach’s alpha levels of 0.73 for the Cohesion subscale and 0.71 for the Adaptability subscale were obtained.

Psychological distress. Psychological distress was assessed using two measures (anxiety and depression). Level of anxiety was assessed using the Trait-Anxiety Scale of the Trait-State Anxiety Inventory for Children (STAIC) [32]. The Trait-Anxiety (TA) scale measures dispositional, or more stable, levels of anxiety. Cronbach’s alpha is 0.85 for TA in the Spanish version [32]. In the present study, reliability was similar (TA, α = 0.88).

The level of depression was assessed using the Children Depression Inventory (CDI) [33], which consists of 27 items and aims to measure the cognitive, affective, and behavioral signs of depression. The Spanish manual reports information on suggested cut-off points for clinical depression (>19). The Spanish version’s internal reliability is 0.69 [33] and in the current study, the reliability index was higher (α = 0.82).

Disordered eating symptomatology. Disordered eating symptomatology was assessed using the level of body esteem and disordered eating attitudes. Body esteem was assessed using the Body Esteem Scale (BES) [34]. It consists of 24 items that measure body esteem in children over 7 years. Cronbach’s alpha is 0.80 for the Spanish version [34]. In the current study, reliability was higher (α = 0.91).

Disordered eating attitudes were assessed using the Children’s Eating Attitudes Test (ChEAT) [34]. It consists of 26 items, which assess eating attitudes and behaviors in children. Children who had tried to lose weight, felt too fat, and thought that their friends would like them more if they were thinner had significantly higher total scores on the ChEAT. The cut-off point for the scale was 20 for the risk of disordered eating. Cronbach’s alpha is 0.76 in the Spanish version [34]. In the present study, reliability was high (α = 0.80).

LOC eating. The presence of episodes of LOC eating was assessed by trained interviewers through a semi-structured interview on the eating habits and eating behavior of the child. The interview was conducted first with the child alone. Subsequently, the information provided by the child was verified by the primary caregiver alone. The presence of LOC episodes was determined following the proposed criteria [4]. For the current study, LOC was defined as not present = 0 and present = 1.

### 2.3. Statistical Analysis

Data were analyzed using the statistical software package SPSS 24.0 for Windows. Data are presented with the mean and standard deviations and percentages were used to describe the categorical variables. BMI standard deviation scores (BMI z-scores) were computed by comparing the children’s BMI with the ideal BMI of the general population of the same sex and age [35]. Two-tailed partial correlations were calculated, controlling for sex, age, and socioeconomic status. The three groups by status of weight or by presenting (or not presenting) LOC episodes were compared using multivariate analysis of covariance, controlling for sex, age, and socioeconomic status. Post-hoc Bonferroni comparisons for continuous variables and chi-square tests for categorical variables were used.

To examine serial multiple mediation models, the PROCESS macro was used. Then, 95% bias-corrected bootstrap confidence intervals were used to judge the significance of the indirect effect, with confidence intervals resampled 10,000 times for each analysis. All *p* values were two-tailed and statistical significance was set at *p* < 0.05.

## 3. Results

### 3.1. Family Environment, Psychological Distress, and Disordered Eating Symptomatology by Weight Status and Their Relationship with Child BMI

Differences in EE, family functioning, depression, anxiety, body esteem, and disordered eating attitudes by weight status are summarized in Table 1, controlling for sex and age of the child and socioeconomic status.

The relationship between children’s BMI and family environment, psychological distress, and disordered eating symptomatology was investigated using partial correlations, controlling for sex, age, and socioeconomic status. Children’s BMI was related to the family’s EE, psychological distress, and disordered eating symptomatology of the child. No significant correlation was found between the child’s BMI and adaptability or cohesion scales.

### 3.2. Differences in Family Environment and Psychological Distress by LOC Eating Episodes

The prevalence of LOC eating episodes in the whole sample was 27.8%, in which 6.6% of the NG, 25% of the OWG, and 52% of the OG had LOC, with significant statistical differences between the groups (χ^2^ = 39.21, *p* < 0.0001). No significant differences between groups were found for sex, age, or socioeconomic status.

For our second aim, the sample was divided into three groups: NG without LOC; OWG/OG without LOC; and OWG/OG with LOC (Table 2). Four children (6.6%) who presented LOC eating in the NG were excluded as analyses with this number were not possible. Demographical variables of age, sex, and socioeconomic status were entered as control variables in all analyses.

### 3.3. Effect of EE on LOC Eating Episodes via Psychological Symptomatology (Anxiety and Depression Models)

Our third aim was to test a model of emotional regulation of LOC eating based on a dysfunctional family environment. Since no significant differences were found in the FACES scales of family functioning in the sample, we choose EE as the family environment variable for the model. Appendix A
Table A1 shows the relationship between all measures. To evaluate the EE as a construct, the cut-off points of the CC and EOI scales were used and ordinal variable high-EE was calculated (0 = non-presence of EE, 1 = presence of EE above or equal of one of the cut-off points of CC or EOI, 2 = presence of EE above or equal of both cut-off points of CC and EOI).

Two paths of the affect regulation model were tested. Statistical procedures outlined by Baron and Kenny (1986) [36] and the significance test of the indirect effect were performed to examine the mediation hypotheses. Demographical variables of age, sex, and socioeconomic status were entered as control variables in all equations.

#### 3.3.1. Depression Model for LOC Eating Episodes

A simple mediation model was tested to examine the mediation role of the level of depression in the presence of LOC eating episodes. There was no significant indirect effect of EE via levels of depression (unstandardized beta [B] = 0.07, standard error [SE] = 0.07, 95% CI: −0.07 to 0.23).

Taking into account the previous literature and the results obtained in the previous section, we tested a second model, including disordered eating attitudes as a second mediator. There was a significant indirect effect of EE via depressive symptomatology and disordered eating attitudes (B[SE] = 0.08[0.05], 95% CI: 0.01 to 0.22) (Figure 1). High-EE predicted higher depressive symptomatology, which predicted higher scores on disordered eating symptoms, which, in turn, predicted LOC eating episodes. At the same time, the relationship between the independent variable (High-EE) and the dependent variable (LOC eating episodes) was still significant when taking the mediators into account (B[SE] = 0.47[.22]; *p* = 0.03) (Appendix A
Table A2). However, the coefficient was smaller than the unmediated coefficient (b = 0.53 vs. 0.80), meaning a significant partial mediation of both mediators between High-EE and LOC eating episodes.

#### 3.3.2. Anxiety Model for LOC Eating Episodes

A simple mediation model was tested to examine the mediation role of the level of anxiety in the presence of LOC eating episodes. There was no significant indirect effect of EE via levels of depression (B[SE] = −0.01[0.01], 95% CI: −0.05 to 0.01). Taking into account the previous literature and the results obtained in the previous section, we tested a second model, including disordered eating attitudes as a second mediator.

A serial multiple mediation model was tested to examine the mediation roles of anxiety (mediator 1) and disordered eating attitudes (mediator 2) in the presence of LOC eating episodes (Figure 2). There was a significant indirect effect of EE via anxiety and disordered eating (B[SE] = 0.04[0.02], 95% CI: 0.03 to 0.11) (Appendix A
Table A1). The direct effect of High-EE on LOC eating episodes, controlling for the mediators, was significant and smaller than the unmediated coefficient (b = 0.47 vs. 0.80). Thus, there was a partial mediation of anxiety and disordered eating attitudes between High-EE and LOC eating episodes (Appendix A
Table A2).

## 4. Discussion

The general aim of this study was to evaluate family environment and their children’s psychological distress based on BMI and the presence of LOC episodes. To the best of our knowledge, this study can be considered the first to compare the family’s EE between these samples.

Our first hypothesis was partially confirmed. In this study, as expected, children’s BMI was related to the family’s EE, psychological distress, and disordered eating symptomatology of the child.

In the literature, aspects such as family conflict, high levels of parental control, and poor communication were associated with higher weight status [14,21,26]. Although parent-child interactions are associated with children’s weight status and appear to play a role in children’s weight regulation [26], the family’s EE has been scarcely studied in terms of obesity. We have not found previous studies that evaluate it in childhood, but our results are similar to those found in the adolescent population with obesity [17]. Its relationship with psychological distress of offspring [12], poorer clinical course, diminished treatment response, and higher rates of relapse [12,13] make it a variable of interest in the study and treatment of obesity.

Regarding family functioning, there was unexpectedly no relationship with children’s BMI, as we had hypothesized. Other studies have found that children from families with lower cohesion tended to present a higher BMI than children with better family cohesion and adaptation [15]. The cohesion and adaptability mean scores in the study sample are similar to those found in the general population in the original scale [37].

As for children’s psychological distress, the presence of a deficit in emotion regulation has been frequently observed in children with obesity [25]. Likewise, children with proper self-regulatory skills are at a lower risk of childhood obesity [26]. Therefore, children’s weight regulation seems to be a specific form of self-regulation [26], emphasizing the importance of studying eating behaviors related to emotion regulation. Regarding disordered eating symptomatology, BMI is associated with lower body esteem and higher levels of pathological eating attitudes. These results emphasize the risk of children with obesity developing eating disorders in their youth, especially considering that these types of variables tend to increase during adolescence [20].

Our second hypothesis was also partially confirmed. Concerning LOC eating episodes, as expected, OG children presented a higher frequency compared to the other two groups. These results present similar prevalence rates as in a previous meta-analysis [5]. The cross-sectional nature of the study does not allow us to identify causality. Since previous studies found that excess weight appears mainly before LOC episodes [38], the relationship will most likely be bidirectional. Children with higher weight status are at increased risk of developing LOC and in turn, the LOC favors a gradual increase in weight [3].

Children with LOC have higher levels of family EE and higher levels of emotional distress than NG. These results reiterate the psychological [3,11] and family [19] discomfort that has been previously related in the literature with the LOC eating construct.

Contrary to what was expected, there were no differences in family environment or psychological distress between OWG/OG children with LOC and without LOC. Both groups showed higher family emotional over involvement and psychological distress than NG children. Despite this, in this study, both EE and psychological distress were related to the presence of LOC eating. Body esteem and disordered eating attitudes seem to be the variables that best differentiate children with LOC eating within the sample with childhood obesity. Due to the relationship between these variables and the development of eating disorders [20], these results reinforce the construct of LOC eating as a possible link between childhood obesity and eating disorders.

It is necessary to note that 13.5% of OWG/OG without LOC scored above the cut-off point in disordered eating attitudes, which implies they are at risk of developing eating problems. Puberty is associated with body image disturbances [24] and furthermore, longitudinal studies show an increase in the prevalence of LOC eating over time [9,10]. Given the age of the sample evaluated, the possibility that psychological distress may be related to the subsequent development of LOC eating should not be overlooked. Another possibility is also contemplated. Other studies have also found no difference in general family communication between children with and without LOC eating [16,23]. However, findings indicate that parents of children with LOC eating express more critical comments explicitly related to their child’s shape, weight, or eating [16], as well as more maladaptive family functioning during family mealtimes [23]. Both factors could favor the appearance of disordered eating attitudes [20,21]. More research would be necessary to identify differences in the content of the criticism from the parents of children with obesity and with LOC eating.

Our third hypothesis was confirmed as the proposed affect regulation model between dysfunctional family environments and LOC was satisfactorily explained by the children’s depression, anxiety, and disordered eating attitudes. In the case of this study, the dimension of dysfunctional family environment was measured only by EE as family functioning did not discriminate between groups. These results are in line with the previous literature [19]. In the same way, both negative affect and eating disorder-specific cognitions about food/eating and body image [39] were identified as antecedents of LOC eating. However, due to the cross-sectional nature of our study, these results are only indicative and must be verified in longitudinal studies.

Other studies have found differences in psychological symptomatology between people who were seeking treatment and those who were not. In this study, participants were recruited during their routine pediatric visits (child checkup and minor health problems), as well as in schools to participate in a study on health habits in the child population. However, the Spanish health system includes a children’s growth annual review at primary medical care, so it was not possible to control the seeking treatment effect, representing a limitation of the study. Since participants with normal-weight and overweight/obesity were recruited from different places (primary school and primary health center, respectively) for this study, it is possible that there may have been unmeasured factors that differentiate these samples. For example, as in the previous limitation, children who regularly are taken to their primary health centers may present psychological problems compared to children who do not regularly go to their pediatrician. Although the results obtained are consistent with the previous literature, this is another limitation of the study.

In conclusion, family difficulties and psychological burdens that children with childhood obesity present are further evidence of the importance of including psychological variables, such as emotional distress or disordered eating symptomatology, in the assessment protocols for obesity in children. Likewise, childhood obesity could benefit from tailored psychological treatments and family interventions [40,41]. The results also highlight the high prevalence of LOC eating in this population. Because of its relationship with disordered eating attitudes and BED, children with this type of disordered eating may not benefit from the usual treatment of obesity (pediatrician or endocrinologist/dietitian). Some of the strategies that are commonly provided to children with obesity, such as reducing the intake of sweets or fats, as well as weight-focused treatments, may even worsen loss of control over eating, such as occurs in eating disorders. For this reason, it may be necessary to assess this disordered eating in the childhood obesity population and work on this aspect in the first place to achieve an effective intervention for the child’s weight status.

## Figures and Tables

**Figure 1 children-07-00225-f001:**
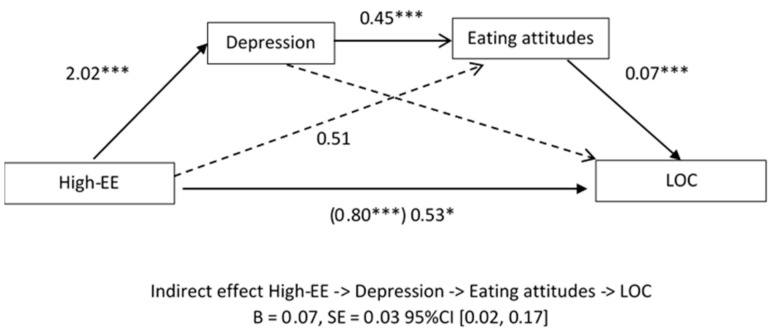
Model of the relationship among depression, eating attitudes, high expressed emotion, and LOC eating episodes for the child and adolescent sample, controlling for sex, age, and socioeconomic status. Values presented are unstandardized regression coefficients. Numbers in parentheses represent standard errors. The value in parentheses represents the coefficient for the direct (unmediated) path. * = *p* < 0.05; *** = *p* < 0.001; High-EE = high expressed emotion; Eating attitudes = disordered eating attitudes; LOC = loss of control eating episodes; B = unstandardized beta; SE = standard error.

**Figure 2 children-07-00225-f002:**
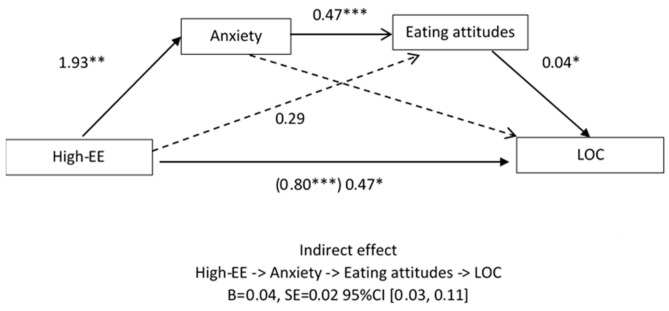
Model of the relationship among anxiety, eating attitudes, high expressed emotion, and LOC eating episodes, controlling for sex, age, and socioeconomic status. Values presented are unstandardized regression coefficients. Numbers in parentheses represent standard errors. The value in parentheses represents the coefficient for the direct (unmediated) path. * = *p* < 0.05; ** = *p* < 0.01; *** = *p* < 0.001 High-EE = high expressed emotion; Eating attitudes = disordered eating attitudes; LOC = loss of control eating episodes; B = unstandardized beta; SE = standard error.

**Table 1 children-07-00225-t001:** Family environment and psychological variables by weight status and their relationship with child BMI, controlling for sex, age, and socioeconomic status.

	NG (*N* = 76) M (SD)	OWG (*N* = 81) M (SD)	OG (*N* = 82) M (SD)	Overall *p*-Value	NG vs. OWG	NG vs. OG	OWG vs. OG	z-BMI Correlation (*p*-Value)
**Expressed emotion**						
Criticism (FQ)	18.37 (0.78)	20.92 (0.70)	20.87 (0.67)	**0.02**	**0.04**	**0.04**	1.00	**0.18 (0.02)**
Overinvolvement (FQ)	20.84 (0.67)	22.99 (0.60)	24.42 (0.58)	**0.001**	0.06	**<0.0001**	0.26	**0.26 (0.001)**
**Patterns of family functioning**								
Adaptability (FACES)	24.01 (0.79)	23.61 (0.71)	23.26 (0.68)	0.78	1.00	1.00	1.00	−0.05 (0.26)
Cohesion (FACES)	39.04 (0.82)	38.26 (0.73)	37.44 (0.70)	0.34	1.00	0.44	1.00	−0.04 (0.52)
**Child Psychological distress**						
Depression (CDI)	6.67 (0.80)	8.41 (0.72)	9.40 (0.69)	**0.04**	0.33	0.03	0.96	**0.16 (0.03)**
Anxiety (STAIC)	30.02 (1.01)	32.10 (0.91)	33.77 (0.87)	**0.02**	0.39	0.01	0.55	**0.20 (0.01)**
**Child disordered eating symptomatology**								
Body Esteem (BES)	19.68 (0.84)	14.10 (0.75)	10.94 (0.72)	**<0.0001**	**<0.0001**	**<0.0001**	**0.009**	**−0.50 (<0.0001)**
Eating attitudes (ChEAT)	5.27 (1.10)	8.07 (0.99)	12.33 (0.95)	**<0.0001**	0.19	**<0.0001**	**0.007**	**0.36 (<0.0001)**

Note: In bold, *p* < 0.05 values for the differences between the groups. BMI = Body Mass Index, M = mean, SD = standard deviation, NG = Normal-weight group, OWG = Overweight group, OG = Obesity Group. FQ = Family Questionnaire; FACES = Family Adaptation and Cohesion Scales; CDI = Children Depression Inventory; STAIC = State-trait anxiety inventory for children; BES = Body Esteem Scale; ChEAT = Children Eating Attitudes Test.

**Table 2 children-07-00225-t002:** Differences between means and standard deviations of the family environment and psychological distress for the NG, OWG/OG without LOC, and OWG/OG with LOC, controlling for socioeconomic status.

	NG without LOC (*N* = 72) M (SD)	OWG/OG without LOC (*N* = 103) M (SD)	OWG/OG with LOC (*N* = 60) M (SD)	Overall *p*-Value	NG without LOC vs. OWG/OG without LOC	NG without LOC vs. OWG/OG with LOC	OWG/OG without LOC vs. OWG/OG with LOC
BMI †	0.15 (0.10)	1.96 (0.08)	2.38 (0.09)	<0.0001	<0.0001	<0.0001	0.003
**Expressed emotion**							
Criticism (FQ)	18.26 (0.83)	20.43 (0.63)	21.65 (0.77)	0.01	0.12	0.01	0.66
Cut-off (%)	16.4	27.8	40.4	0.01	0.09	0.003	0.11
Overinvolvement (FQ)	20.72 (0.72)	23.41 (0.54)	24.03 (0.67)	0.003	0.01	0.003	1.00
Cut-off (%)	8.8	24.7	37.9	0.001	0.009	<0.0001	0.08
**Patterns of family functioning**							
Adaptability (FACES)	24.34 (0.83)	23.39 (0.63)	23.42 (0.77)	0.63	1.00	1.00	1.00
Cohesion (FACES)	39.06 (0.87)	38.13 (0.66)	37.50 (0.80)	0.43	1.00	0.59	1.00
**Child Psychological Distress**							
Depression (CDI)	6.50 (0.85)	8.68 (0.64)	9.32 (0.79)	0.04	0.13	0.05	1.00
Cut-off (%)	0	8	7.1	0.05	0.01	0.02	0.84
Anxiety (STAIC)	29.26 (1.06)	32.51 (0.80)	34.16 (0.98)	0.004	0.05	0.003	0.58
**Child disordered eating symptomatology**							
Body Esteem (BES)	19.88 (0.89)	13.70 (0.67)	10.70 (0.82)	<0.0001	<0.0001	<0.0001	0.01
Eating attitudes (ChEAT)	5.05 (1.17)	8.78 (0.88)	12.74 (1.08)	<0.0001	0.04	<0.0001	0.01
Cut-off (%)	3.8	13.5	21.4	0.02	0.05	0.006	0.02

Note: In bold, *p* < 0.05 values for differences between the groups. † BMI standard deviation scores were computed by comparing the subjects body mass index with the ideal BMI of the general population of the same sex and age [35]. BMI = Body mass index, NG = Normal-weight group, OWG/OG = Overweight and obesity group, LOC = Loss of control eating episodes, FQ = Family Questionnaire; FACES = Family Adaptation and Cohesion Scales; CDI = Children Depression Inventory; STAIC = State-trait anxiety inventory for children; BES = Body Esteem Scale; ChEAT = Children Eating Attitudes Test.

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
