# Peer review of "Identifying Loss of Control Eating within Childhood Obesity: The Importance of Family Environment and Child Psychological Distress"

_children, 2020, doi:10.3390/children7110225_

Round 1

Reviewer 1 Report

Thank you for the opportunity to review this manuscript. This is an interesting topic that can be considered by readers. However, this manuscript has some concerns that make it inappropriate for publication and should be addressed before publication. The following are suggested for improvement:

Abstract:

 (a) The authors should specify which variables fall within the categories "family functioning”, "psychological functioning" etc.  because some variables suddenly appear in the Results section ("expressed emotion", "depression / anxiety" etc.)  and it is not known which category these variables refer to. I think it would be good to unify this description with the aims of the study presented in the Introduction section.

(b) “Children with obesity and 22 LOC presented higher body mass index, worse body esteem, and disordered eating attitudes.” than?

Abstract and other section of manuscript:

(a) It should be considered whether these terms are synonyms: disordered eating and eating disorder (e.g. https://www.psychologytoday.com/us/blog/contemporary-psychoanalysis-in-action/201402/disordered-eating-or-eating-disorder-what-s-the). In my opinion, there are significant differences between these categories.

(b) The description of the main objectives of the study should be unified in the Abstract and Introduction sections.

(c) “Children with overweight/obesity, both with or without LOC, showed higher psychological distress and expressed emotion than normal-weight children” <- This sentence is not entirely consistent with the results in Table 2.

Introduction:

(a) Please clarify:

* Are the group pre-adolescents? We can find information about age in the Participants and procedure section (“The sample consisted of 239 Spanish families of children aged 8 to 12…”). In my opinion, pre-adolescence period involves a smaller age range (e.g. about 12).

* “LOC eating episodes will be more frequent in children with OWG/OG” …than? (NG?)

* The authors analysed the differences between the groups (and not the relationship of the LOC with other variables), therefore maybe “the children with LOC” -> “LOC eating will be associated with a more highly dysfunctional environment, higher psychological distress, and higher ED symptomatology than in the children without LOC eating.”

Results:

(a) The hypothesis (“(c) the psychological distress (anxiety and depression) of the children will mediate the relationship between dysfunctional family environments and LOC eating episodes.”) presented by the authors indicates that 1 mediator (depression/anxiety) will be analysed and the authors will use the model 4 twice.

In the results section, it suddenly turns out that they analyse the influence of two mediators simultaneously (model 6; 2 mediators: depression/anxiety and eating attitudes).

Discussion:

(a) To a small extent the authors discuss their data in a manner that provides insight beyond that presented in previous sections.

(b) To a small extent the authors make suggestions as to how the results of their study need to be extended in the future to learn more about the issue in question.

(c) I would recommend expanding the Discussion section as to how some of the results of the study could be implemented rather than a repeat of what was already discussed. Or perhaps discuss how there may be barriers to implementing the results of study.

Author Response

We would like to thank you for the comments and suggestions. We are pleased to have the opportunity to contribute to an upcoming issue of Children. Please see below for the details regarding how we have addressed each of the concerns raised. New changes have been highlighted in yellow in the final version of the manuscript.

ABSTRACT

(a) The authors should specify which variables fall within the categories "family functioning”, "psychological functioning" etc.  because some variables suddenly appear in the Results section ("expressed emotion", "depression / anxiety" etc.)  and it is not known which category these variables refer to. I think it would be good to unify this description with the aims of the study presented in the Introduction section.

- Following the recommendation, we have specified the variables measured within the categories: family environment, psychological distress and disordered eating symptomatology, unifying them with the aims of the study. To facilitate understanding, the term "family functioning" has also been changed to "family adaptability and cohesion" in the aims of the study.

(b) “Children with obesity and 22 LOC presented higher body mass index, worse body esteem, and disordered eating attitudes.” than?

-  We have specified in the Abstract section that children with obesity and LOC presented higher body mass index, worse body esteem, and disordered eating attitudes than children without LOC.

ABSTRACT AND OTHER SECTION OF MANUSCRIPT

(a) It should be considered whether these terms are synonyms: disordered eating and eating disorder (e.g. https://www.psychologytoday.com/us/blog/contemporary-psychoanalysis-in-action/201402/disordered-eating-or-eating-disorder-what-s-the). In my opinion, there are significant differences between these categories.

Thank you for your comment. We agree that disordered eating and eating disorders are different categories. In this manuscript, both terms are used to refer to different aspects:

- LOC eating is currently considered as disordered eating, as it does not meet the currently established criteria for a binge eating disorder, as the amounts of food eaten in the episodes are not objectively large.

- The ChEAT questionnaire measures disordered eating symptoms in the child. A score of over 20 is considered a cut-off point of high risk of an eating disorder.

- We have changed the expression “eating disorder symptomatology” to “disordered eating symptomatology” for consistency in all the ms.

 (b) The description of the main objectives of the study should be unified in the Abstract and Introduction sections.

The description of the main objectives of the study has been unified in the Abstract and Background sections.

(c) “Children with overweight/obesity, both with or without LOC, showed higher psychological distress and expressed emotion than normal-weight children” <- This sentence is not entirely consistent with the results in Table 2.

 - Thank you for bringing this concern to our attention. The results in Table 2 show higher levels of anxiety, higher frequency of scores above the cut-off of depression and higher levels of emotional overinvolvement. We have changed the sentence in the Abstract and Discussion section to be consistent with the results.

INTRODUCTION

(a) Please clarify:

* Are the group pre-adolescents? We can find information about age in the Participants and procedure section (“The sample consisted of 239 Spanish families of children aged 8 to 12…”). In my opinion, pre-adolescence period involves a smaller age range (e.g. about 12).

- The age range for pre-adolescence is commonly designated as 10-13 years, but the American Academy of Pediatrics notes that changes begin to occur between the ages of 8 and 9 (https://www.healthychildren.org/English/ages-stages/teen/Pages/Stages-of-Adolescence.aspx). To avoid confusion, we have replaced “pre-adolescents” with “children” in the objectives of the study section.

* “LOC eating episodes will be more frequent in children with OWG/OG” …than? (NG?)

- It has been specified in the hypotheses section that LOC eating episodes will be more frequent in children with OWG/OG than in NG.

* The authors analysed the differences between the groups (and not the relationship of the LOC with other variables), therefore maybe “the children with LOC” -> “LOC eating will be associated with a more highly dysfunctional environment, higher psychological distress, and higher ED symptomatology than in the children without LOC eating.”

 - Following the recommendation, we have changed “LOC eating” to “children with LOC eating” in the hypotheses section.

RESULTS

(a) The hypothesis (“(c) the psychological distress (anxiety and depression) of the children will mediate the relationship between dysfunctional family environments and LOC eating episodes.”) presented by the authors indicates that 1 mediator (depression/anxiety) will be analysed and the authors will use the model 4 twice. In the results section, it suddenly turns out that they analyse the influence of two mediators simultaneously (model 6; 2 mediators: depression/anxiety and eating attitudes).

- Our initial objective was to develop two models, one with the depression variable and the other with the anxiety variable as mediator. It has been described in the hypotheses section that anxiety and depression will be evaluated separately.  As explained in the Results section, these first models did not obtain significant results. Taking into account the previous literature and the results obtained in the previous section, we tested other two models, including disordered eating attitudes as a second mediator. For this reason, four models have been used in the study.

It has been clarified in the objectives and results sections (line 100, 248-252).

DISCUSSION

(a) To a small extent the authors discuss their data in a manner that provides insight beyond that presented in previous sections. (b) To a small extent the authors make suggestions as to how the results of their study need to be extended in the future to learn more about the issue in question. (c) I would recommend expanding the Discussion section as to how some of the results of the study could be implemented rather than a repeat of what was already discussed. Or perhaps discuss how there may be barriers to implementing the results of study.

- Following the recommendation, the Discussion section has been expanded providing information about how the results of the study can be implemented (line 345-356).

Reviewer 2 Report

This manuscript examined associations between child BMI and LOC eating with family and psychological functioning. They found higher expressed emotion, anxiety, and depression in the children with obesity and overweight compared to normal weight. They also found LOC to be more common with increasing BMI. Children with OW/OB and LOC had higher EE, depression, anxiety, lower body esteem, and more disordered eating attitudes than children without LOC who were normal weight. Finally, they found that depression, anxiety, and disordered eating attitudes mediated the relationship between EE and LOC eating. Overall I found this study to be interesting, studying EE in the context of childhood obesity and LOC eating which hasn't been examined much. My comments are detailed below.

Abstract

  • Line 20-21: add that assessment was done in children attending PCP appointment and their caregiver
  • Line 22-23: children with LOC and OW/OB compared to whom?

Introduction

  • Line 62: Clearer definition of expressed emotion would be helpful since this is a primary construct of the study. I don't think "emotional experience of caring for a family member" is an accurate description. 

Methods

  • More detail is needed about study procedures. Who completed which questionnaire (parent vs child)? Is there information about which caregiver completed the questionnaires? Were children given privacy from parents when completing their questionnaires? Where were the surveys done? Mailed to the home? Online? Done in the doctor's office?
  • When assessing LOC eating, who did the semi-structured interview? Was it in person? What was the time frame in which LOC eating was assessed?
  • Were participants paid for participation?
  • What was the reason for recruiting normal weight participants differently than overweight/obese? Why not recruit all from the doctor's offices? I worry that, because the analyses specifically compare these groups, there could be other unmeasured factors that are different about the samples that might explain some of the results. 

Results

  • In Table 1, it would be helpful to label the family environment subscales with their overall construct (so reminding readers that criticism and overinvolvement are subscales of EE)
  • Abbreviation for weight-based groups is inconsistent (for obesity it's sometimes OB, sometimes OG and for overweight it's sometimes OWG, sometimes OW). 
  • Line 194 - language is confusing. I think authors mean that 6.6% of the NWG had LOC, 25% of the OWG had LOC, and 52% of the OG had LOC, but this is not how it is written. Please clarify.
  • Line 220 - says that depression during LOC is measured - I don't think this is what was measured (rather depression in general)

Discussion

-Line 271: no relationship with...

-Line 292: clarify that EE is in the family, not in the child

-Line 312-217: rather than using abbreviation CC, it would be clearer to say criticism

Author Response

We would like to thank you for the comments and suggestions. We are pleased to have the opportunity to contribute to an upcoming issue of Children. Please see below for the details regarding how we have addressed each of the concerns raised. New changes have been highlighted in yellow in the final version of the manuscript

ABSTRACT

Line 20-21: add that assessment was done in children attending PCP appointment and their caregiver

- We have included this information in the Abstract section.

Line 22-23: children with LOC and OW/OB compared to whom?

- We have described in the Abstract that children with obesity and LOC presented higher body mass index, worse body esteem, and disordered eating attitudes than children without LOC.

INTRODUCTION

Line 62: Clearer definition of expressed emotion would be helpful since this is a primary construct of the study. I don't think "emotional experience of caring for a family member" is an accurate description. 

-  We have included a more extensive definition of expressed emotion (line 60-65).

METHODS

More detail is needed about study procedures. Who completed which questionnaire (parent vs child)? Is there information about which caregiver completed the questionnaires? Were children given privacy from parents when completing their questionnaires? Where were the surveys done? Mailed to the home? Online? Done in the doctor's office?

- Participants were summoned to their Primary Care Center or their primary school, where a trained interviewer conducted a semi-structured interview, with the primary caregiver and the child separately, and provided the questionnaires. The questionnaires were completed at the time of assessment. Family environment questionnaires (FQ and FACES) were answered by the primary caregiver, while the remaining questionnaires (CDI, STAIC, BES and ChEAT) were answered by the child. Participants' privacy was respected, and both the primary caregiver and the child answered the questionnaires alone. This information has been included in the procedure section (line 124-131).

When assessing LOC eating, who did the semi-structured interview? Was it in person? What was the time frame in which LOC eating was assessed?´

- The presence of episodes of LOC eating was assessed in person by trained interviewers through a semi-structured interview on the eating habits and eating behavior of the child. The interview was conducted first with the child alone. Subsequently, the information provided by the child was verified by the main caregiver alone. The time frame was about 20 minutes.

We have included this information in the instruments of assessment section (line 166-173).

Were participants paid for participation?

- Participation in the study was voluntary and participants did not receive compensation. We have included this detail in the Participants and Procedure section (line 123)

What was the reason for recruiting normal weight participants differently than overweight/obese? Why not recruit all from the doctor's offices? I worry that, because the analyses specifically compare these groups, there could be other unmeasured factors that are different about the samples that might explain some of the results. 

- Normal-weight participants were recruited from primary schools to obtain a larger sample. In an attempt to improve the external validity of the study, we chose “Goya” and “Daroca” Health Center because these centers concentrate the highest percentage of overweight children in the region of Madrid. However, our sample may not be representative of this target population. If we question how our sample would differ from an ideal sample that was randomly selected, we think that the main difference would be the intrinsic interest in participating and its associated variables. We agree that this may influence the results, since children with obesity who attend the Primary Care Center may have more difficulties than those who do not. To reduce the possible effect of unmeasured factors as much as possible, we decided to separate the sample with overweight and obesity collected in Primary Care Centers and evaluate the differences between them (Table 1). Since differences have been found, and the results are consistent with the previous literature, we assume that at least a part of the differences that occur with children with normal weight are due to variables related to weight status.

For our second objective, the differences between the samples recruited are less relevant, since the main objective is to find differences between overweight / obese children who have LOC and those who do not.

Despite this, following the recommendation, we have included in the limitations section the possible presence of unmeasured factors in the recruitment differences between the samples.

RESULTS

In Table 1, it would be helpful to label the family environment subscales with their overall construct (so reminding readers that criticism and overinvolvement are subscales of EE)

- Following the recommendation, we have described in Table 1 and 2 both the expressed emotion construct and family functioning patterns.

Abbreviation for weight-based groups is inconsistent (for obesity it's sometimes OB, sometimes OG and for overweight it's sometimes OWG, sometimes OW). 

- We apologize for these misprints in the text, they have been corrected following the abbreviations indicated in the objectives section (OWG and OG).

Line 194 - language is confusing. I think authors mean that 6.6% of the NWG had LOC, 25% of the OWG had LOC, and 52% of the OG had LOC, but this is not how it is written. Please clarify.

-  Following the recommendation, we have modified that sentence to make it clearer.

Line 220 - says that depression during LOC is measured - I don't think this is what was measured (rather depression in general)

- We have changed the expression the mediation role of depression "during LOC eating episodes" for "in the presence of LOC eating episodes".

DISCUSSION

-Line 271: no relationship with...

- We have described in line 281 that no relationship is found with the child's BMI status.

-Line 292: clarify that EE is in the family, not in the child.

- We have clarified that EE is in the family.

-Line 312-217: rather than using abbreviation CC, it would be clearer to say criticism

- Following the comment, we have changed CC for critical comments (line 319) and criticism (line 323).

Reviewer 3 Report

Thankyou for the opportunity to review your manuscript.

I have a few suggestions:

Throughout:

Is 'poorer' a more appropriate word to describe poor body esteem than 'worse'?

Abstract:

Line 27 show could be changed to exhibit

Line 28: the word 'issues' or 'poorer' needs to be added to body esteem

Line 29: deal could be changed to cope

Introduction:

Expressed emotion needs to be more fully descibed, particularly its potential negative impacts.

Line 76: no man's land; please change to academic language. For example, role of LOC in childhood obesity is yet to be fully understood/described.

Methods:

Was there only one child per family? Were families with only one child eligible to participate?

Please describe who performed the measurements and who completed the questionnaires (researchers? parents on behalf of their children? children?).

How much missing data were there? How was this managed during analysis?

Please add detail about the semi structured interview.

Discussion:

Please define CC.

Please consider adding a sentence or two related to limitations (differences in sampling approaches, risk of bias).

Author Response

We would like to thank you for the comments and suggestions. We are pleased to have the opportunity to contribute to an upcoming issue of Children. Please see below for the details regarding how we have addressed each of the concerns raised. New changes have been highlighted in yellow in the final version of the manuscript

Is 'poorer' a more appropriate word to describe poor body esteem than 'worse'?

-  Following the recommendation, the word “worse” has been changed to “poorer”.

ABSTRACT:

Line 27 show could be changed to exhibit

-  Following the comment, the word “show” has been changed to “exhibit”.

Line 28: the word 'issues' or 'poorer' needs to be added to body esteem

- We apologize for the misprint, "body esteem issues" has been included for consistency.

Line 29: deal could be changed to cope

- Following the recommendation, the word “deal” has been changed to “cope”.

INTRODUCTION:

Expressed emotion needs to be more fully described, particularly its potential negative impacts.

- Thank you for your recommendation. Expressed emotion has been clarified in the Introduction section (line 59-65).

Line 76: no man's land; please change to academic language. For example, role of LOC in childhood obesity is yet to be fully understood/described.

- Following the comment, this sentence has been modified.

METHODS:

Was there only one child per family? Were families with only one child eligible to participate?

- Only one child per family was assessed. However, there were no exclusion criteria based on the number of children in the family.

Please describe who performed the measurements and who completed the questionnaires (researchers? parents on behalf of their children? children?).

- Participants were summoned to their Primary Care Center or their primary school, where a trained interviewer conducted a semi-structured interview, with the main caregiver and the child separately, and provided the questionnaires. The questionnaires were completed at the time of assessment. Family environment questionnaires (FQ and FACES) were answered by the main caregiver, while the remaining questionnaires (CDI, STAIC, BES and ChEAT) were answered by the child. Participants' privacy was respected, and both the primary caregiver and the child answered the questionnaires alone. This information has been included in the procedure section (line 124-131)

How much missing data were there? How was this managed during analysis?

- Missing data was between 7% and 9%. We considered that missing values were missing completely at random (for example, marking multiple answers in a questionnaire). The method used was a complete case analysis, using only the data of variables after removing all missing values.

Please add detail about the semi structured interview.

The presence of episodes of LOC eating was assessed in person by trained interviews through a semi-structured interview on the eating habits and eating behavior of the child. The interview was conducted first with the child alone. Subsequently, the information provided by the child was verified by the primary caregiver alone. The interview lasted around 20 minutes. This information has been included in the procedure section (line 166-173).

DISCUSSION:

Please define CC.

- Following the comment, we have defined CC for critical comments (line 319) or criticism (line 323) by the primary caregivers.

Please consider adding a sentence or two related to limitations (differences in sampling approaches, risk of bias).

- The limitations section has been expanded (line 337-343) taking into account the differences in the sample recruitment. Since participants with normal weight and overweight / obesity have been recruited from different places (primary school and primary health center, respectively), it is possible that there are unmeasured factors that differentiate these samples. For example, in the same way as in the previous limitation, children who go to the primary health center could present more psychological problems than children who do not go to their pediatrician. Although the results obtained are consistent with the previous literature, this is a limitation of the study.

Round 2

Reviewer 1 Report

Thank you for the opportunity to review a revised manuscript. I appreciate the authors' effort in responding to all comments, and I believe that most of the issues have been appropriately addressed.